# Impact of different landing heights on the contact force in the medial tibiofemoral compartment and the surrounding muscle force characteristics in drop jumps

**Gengchao Bi** [1]☯, **Lijun Hua** [2]☯*, **Jiajie Sun** [1], **Qiang Xu** [1], **Guanbo Li** [1]

1 Graduate School, Harbin Sport University, Harbin, Heilongjiang, China, 2 College of Physical Education and Training, Harbin Sport University, Harbin, Heilongjiang, China

☯ These authors contributed equally to this work.
* hualijun@hrbipe.edu.cn

**Data Availability Statement:** The data underlying the findings of this study are freely available at https://doi.org/10.6084/m9.figshare.25953808.v3.

## Abstract

This study explored the impact of landing height on the tibiofemoral joint's medial compartment force (MCF) during drop jumps to help athletes prevent knee injury. Experienced male participants (N = 16) performed drop jumps with landing heights from 0.15 m to 0.75 m. Kinematic/kinetic parameters were collected using a motion capture system and a three-dimensional force platform. The Med-Lat Knee model was used to calculate biomechanical indicators of the knee joint, and data were analyzed using one-way analysis of variance and one-dimensional statistical parametric mapping (SPM1d). Findings indicated that landing height significantly affected the anterior-posterior and vertical MCF, flexion-extension torque, internal-external rotation torque, and vertical ground reaction force (p<0.05)—all increasing with elevated landing height—and significantly impacted the generated force of the vastus medialis, vastus lateralis, and vastus intermedius (p<0.05). SPM1d analysis confirmed these results within specific time intervals. Thus, both the knee moment and the MCF exhibited similar coordinated changes during drop jumps, indicating that these may be adaptive movement strategy. The impact of varying drop jump heights on muscle groups around the knee joint varied suggests that different heights induce specific muscular responses and improve muscle coordination to prevent knee joint injuries.

## 1 Introduction

In a drop jump (DJ), an athlete drops from a given height, transforming the body's weight into momentum to immediately execute a vertical jump as soon as the ground is contacted. This process forces the lower limb muscles to undergo a stretching-shortening cycle to generate rapid strength [1]. The knee joint experiences multi-axial and multi-faceted load pressures, and this complex state of stress can easily lead to abnormal knee joint loading and increase the risk of knee injuries [2, 3]. An imbalance in the mechanical load of the medial compartment of the tibiofemoral joint is one of the primary intrinsic factors leading to osteoarthritis [4].

**Funding:** LH received funding from the Talent Introduction Scientific Research Startup Fund Project of Harbin Sport University (HSU: https://www.hrbipe.edu.cn/)[RC21-202205].The funders had no role in study design, data collection and analysis, decision to publish, or preparation of the manuscript.

**Competing interests:** The authors have declared that no competing interests exist.

Although the knee adduction moment (KAM) has been used as a key mechanical parameter in assessing knee joint loading, its capability in predicting the distribution characteristics of the medial compartment load is limited. In contrast, the contact force in the tibiofemoral compartment can more accurately assess the load distribution characteristics of the knee joint's medial compartment [5, 6]. This contact force, transmitted between the medial and lateral compartments composed of the medial and lateral condyles of the femur and the tibial plateau, is an essential component of the knee joint load [7, 8]. Despite its importance, research specifically focusing on the tibiofemoral joint's medial compartment force (MCF) during DJ movements is scarce. Previous studies have often relied on the KAM as a proxy, which does not fully capture the load distribution characteristics of the medial compartment.

With the development of personalized musculoskeletal modeling technology, the use of static optimization algorithms in musculoskeletal model systems to obtain joint contact forces is more economical and convenient than traditional remote sensing systems [9]. This method has also been validated for its accuracy and scientific nature in calculating contact forces [10]. Among open-source musculoskeletal modeling systems, the Med-Lat Knee model developed by Lerner can precisely measure the MCF and quantitatively analyze the distribution and amplitude changes of these forces [11]. Using this model, one can accurately estimate the intercondylar contact force of the human body during DJ movements, helping to compare the impact of different landing heights on the characteristics of the MCF during DJ movements.

The force generated by muscle contraction can regulate the mechanical load borne by the knee joint. During a DJ, the medial compartment of the knee joint undertakes a significant load, and the stabilizing action of the quadriceps, hamstrings, and gastrocnemius muscles helps balance this load [12, 13]. The influence of this muscle force on joint contact forces, including the quadriceps and hamstrings [14], collectively affects the MCF [15]. At higher drop jump heights, the increased impact forces can lead to changes in muscle activation patterns, resulting in a decrease in knee stiffness [16, 17]. This reduced stiffness could change the load distribution in the MCF, further affecting the tibial shear force and knee joint load [18, 19]. Previous research has shown that external factors such as land height can affect the distribution of knee joint loads by altering the knee joint flexion angle, torque, ligament tension, and muscle contraction force, as well as the stability of the knee joint [20, 21].

The present study utilized the Med-Lat Knee model to calculate and assess the MCF, KAM, and changes in muscle force characteristics during DJ movements at different landing heights. By analyzing the impact of different landing heights on these characteristics, this study aims to provide a more precise understanding of knee joint loading and the role of muscle forces during DJs. This will offer valuable insights for athletes in selecting suitable DJ landing heights to minimize knee joint load, enhancing current knowledge on DJ training and its implications for knee joint health. We assessed the following hypotheses in this study: (1) An increase in the DJ landing height will lead to a significant increase in the MCF; and (2) A higher landing height will significantly increase the KAM and alter the muscle force pattern, leading to a measurable increase in knee joint mechanical load compared to lower landing heights.

## 2 Materials and methods

### 2.1 Subject selection

We recruited 16 male participants with 3–5 years of sports training experience and no lower limb injuries in the past year. A prediction for the sample size was carried out using GPower 3.1.9.7, with a significance level ($\alpha$) of 0.05 and a statistical power of 80%. The analysis determined that a minimum sample size required was 13 individuals. The participants predominantly engaged in sports such as basketball, volleyball, and soccer, which necessitate frequent

jumping and landing movements analogous to the drop jumps executed in this study. Their demographic information is as follows: mean age 23.25±1.04 years, mean body weight 70.01 ±5.77 kg, and mean height 1.77±0.05 m, and they were all right-leg dominant. The recruitment of participants for this study took place between October 23, 2023, and November 5, 2023. Before the start of the experiment, all participants were informed about the procedure and content of the study and provided informed written consent. Participants' consent forms were in writing and were asked to sign before the experiment began. The implementation of this study obtained approval from the Ethics Committee of Harbin Sport University. The approval number is 2024012.

## 2.2 Experimental procedure

The experiment employed eight 600 series high-speed cameras (200 Hz, Qualisys, Sweden), with two cameras mounted on each of the laboratory's four walls at an equal elevation.one 40 cm × 60 cm multi-axis biomechanics force plate (1000Hz, model, BP4000600; AMTI, USA) and A 16-channel Trigno® wireless surface electromyography (EMG) system (2000Hz, DELSYS, USA). The devices were connected through the Qualisys indoor/outdoor motion capture and analysis system to collect kinematic and kinetic parameters of the participants. The synchronous collection of kinematic and kinetic data was completed by Qualisys Track Manager software. The application and location of the markers was primarily based on the study by Delp et al. [22]. (shown in Fig 1). Before applying the reflective markers, the target muscle areas were wiped with alcohol to remove dry skin and oil from the skin surface. If leg hair obstructed the area, it was removed to avoid affecting the experimental results. Afterwards, activated Trigno sensors were attached along the direction of the muscle fibers of the target muscles on the participant's right lower limb using the double-sided adhesive provided with the surface electromyography (EMG) sensors. The specific muscles included were the rectus femoris (RF), vastus medialis (VM), vastus lateralis (VL), semitendinosus (ST), semimembranosus (SM), and biceps femoris (BF) (shown in Fig 1). To ensure consistency of experimental conditions and simulate the actual sports

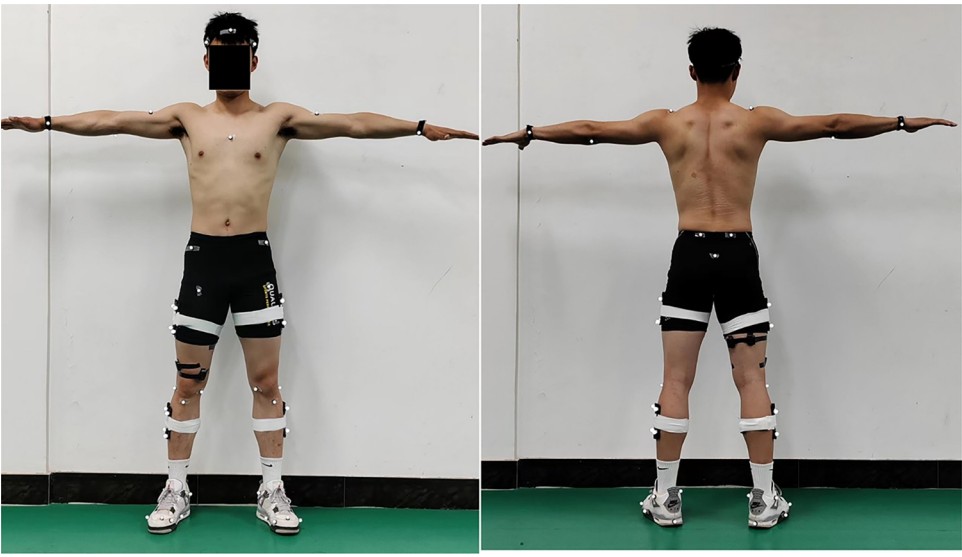

**Fig 1. Specific locations of reflective markers and surface electromyography Sensors.** The left image is a front view; the right image, a rear view.

environment, all participants wore similar sports shoes for the tests. This also helped to minimize the variable effects caused by differences in footwear, while ensuring participants' safety and comfort. After all preparations were completed, each participant warmed up and then sequentially performed single-leg drops and rebound DJs from landing heights of 0.15 m, 0.30 m, 0.45 m, 0.60 m, and 0.75 m, three times at each height, with a 30-s interval between each jump. The standard procedure followed was to stand ready on the jump box, step forward with one foot, lean forward, then drop to the force platform without initial velocity, followed by a rapid vertical jump upward. For all participants, the upper limbs were required to swing naturally with the DJ movements.

### 2.3 Med-Lat Knee model

This study employed a novel Med-Lat Knee full-body musculoskeletal model, a modification of the Gait2392_Simbody model [23]. The Med-Lat Knee model included 18 body segments and 92 muscle actuators, in particular, components of the distal femur and tibial platform [11]. Using OpenSim simulation software, the forces of these muscles were calculated. The JointReaction tool of OpenSim also allowed for the assessment of variations in the MCF joint in three dimensions [24, 25]. Additionally, skeletal simulation software was used to evaluate the knee joint's torque to explore the correlation between the KAM and the MCF [26].

### 2.4 Essential data processing workflow and primary data processing

First, the captured motion data were converted and processed by applying low-pass filters of 10 Hz and 100 Hz to the kinematic and kinetic parameters, respectively. Then, motion simulation was performed using a full-body model equipped with the Med-Lat Knee. The OpenSim simulation workflow included the following steps: (1) Scaling the dynamic model and static kinematic data to match the dimensions of each participant to obtain a scaled model, simultaneously scaling the position of the medial compartment contact points uniformly. (2) Applying dynamic kinematic data to the scaled model for inverse kinematic operations to obtain kinematic parameters, such as joint angles. (3) Applying kinematic data and force platform data to the scaled model to perform inverse dynamic analysis to obtain kinetic data, such as joint moments. (4) Using the static optimization feature to calculate the activation states and force values of the model's muscles. (5) Employing the Joint Reaction function in the Analyze tool, combining inverse kinematics and dynamics results, to calculate knee joint contact forces [22, 27]. Second, the raw surface EMG signals were filtered using a Butterworth filter, followed by full-wave rectification. The process used for handling raw surface EMG signals is summarized as follows: filtering, by applying 20–500 Hz Butterworth band-pass and 50 Hz notch filters; rectification and inversion, by processing the filtered signals using the abs function; normalization, by performing maximum value normalization; feature value calculation, by calculating the root mean square (RMS) values; and plotting the RMS envelope by smoothing the normalized RMS values using the envelope function and a sliding window (50-ms window value) to form the muscle activation curve.

### 2.5 Biomechanical data analysis

We categorized the DJ action into pre-activation, buffering, and push-off phases by measuring the vertical axis force (Fz) on the multi-axis force platform and the position of the body's center of gravity on the vertical axis. The pre-activation phase was defined as 150 ms before the feet contacted the force platform with the Fz value exceeding 5 N [28]. The buffering phase started at the moment of landing and continued until the body's center of gravity reached the lowest point on the vertical axis. The push-off phase began from the lowest point of the body's

center of gravity and continued until the Fz value dropped below 5 N and the foot left the ground. The main research indicators included the following: kinetic indicators of the vertical ground reaction force (vGRF), knee flexion moment (KFM), knee extensor moment (KEM), KAM, and knee rotation moment (KRM), and the contact forces of the anterior-posterior (X), axial (Y), and medial-lateral (Z) directions of the medial tibiofemoral compartment ($MCF_{fx}$, $MCF_{fy}$, and $MCF_{fz}$). The muscle force indicators included the force values of the RF, VM, vastus intermedius (VI), VL, the long head of the biceps femoris (BFlh), and the short head of the biceps femoris (BFsh).

## 2.6 Statistical analysis

This study conducted descriptive statistical analysis using IBM SPSS Statistics 24, analyzing the lower extremity biomechanical indicators for five different heights of DJ actions, with the results presented as the mean ± standard deviation. Based on those feature value results, we further applied one-way analysis of variance (ANOVA) to explore significant inter-group differences. Before conducting an ANOVA, the data were assessed to determine whether they were normally distributed and checked for homogeneity of variances. Multiple comparison tests were used only when the ANOVA results were statistically significant. We used the Bonferroni method for equal variances and Tamhane's T2 method for unequal variances, with the significance level set at 0.05. In addition, we calculated the effect size $\eta^2$ to assess the practical significance of the study results. For one-dimensional curve data, we used Origin 2018 64-bit software for spatiotemporal normalization of the knee joint kinematics, dynamics data, medial compartment contact forces, and muscle force data for five different landing heights of the DJs. The original data were interpolated to 101 data points, with 0% representing 150 ms before landing and 100% corresponding to the moment after landing when the foot left the ground. Based on this, we used F-tests in one-dimensional statistical parametric mapping (SPM1d) for analysis, with a two-tailed test at a significance level set at 0.05, aiming to analyze the statistical differences in continuous data, especially the differences between two sets of data during the movement cycle [29]. All SPM1d analyses were completed in MATLAB 2021a.

## 3 Results

To verify the accuracy of the simulation, we compared the surface EMG signals with the simulation results (Fig 2). One participant was selected to perform the same action to record the original surface EMG signals and the kinematic and kinetic data. After preprocessing the surface EMG signals, the RMS was transformed to a muscle activation state indicator (0 indicating no activation) using the envelope function (SEMG). The kinematic and kinetic data were processed using the OpenSim Static Optimization tool to obtain muscle activation data (Activation). A comparison of these two sets of results indicated that the resulting curves, including those for the BF, VM, VL, and RF, showed good consistency.

As shown in Table 1, the univariate analysis of the knee joint mechanical indicators during the DJs from different landing heights indicated some important findings. First, the landing height significantly affected the vGRF (p = 0.007). Specifically, the vGRF at a height of 0.15 m was significantly lower than that at other heights, while the forces at 0.45 m, 0.60 m, and 0.75 m were significantly higher than that at 0.30 m.

Landing height significantly affected the KFM (p = 0.001), with the moment at 0.45 m, 0.60 m, and 0.75 m being significantly higher than that at 0.15 m and 0.30 m. The landing height also significantly affected the KRM (p<0.01), with the moment at 0.45 m, 0.60 m, and 0.75 m being significantly higher than that at 0.15 m and 0.30 m. Regarding the MCF, the landing height had a significant impact on both the anterior-posterior and vertical contact forces

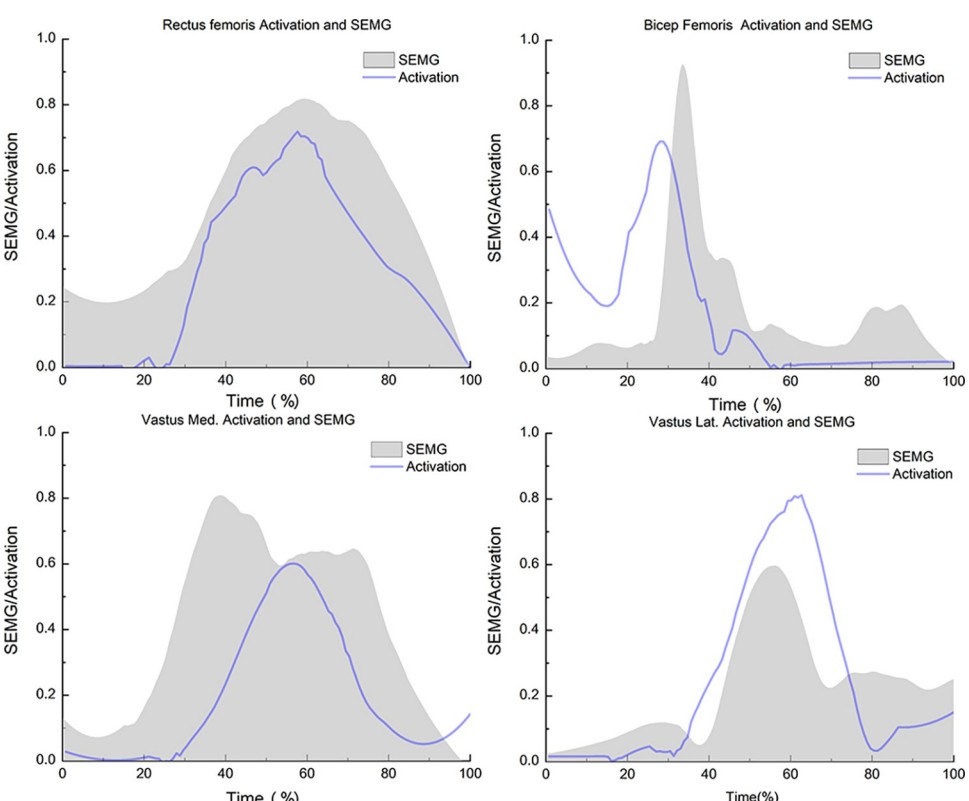

**Fig 2. Comparison of muscle activation estimated by static optimization of the OpenSim and electromyography signals.** Notes: "SEMG" is indicative of the processed data, whereas "Activation" denotes the outcomes of muscle activation following static optimization processing within OpenSim.; Vastus Med., vastus medialis; and Vastus Lat., vastus lateralis.

(p = 0.01 and p<0.01, respectively). At 0.60 m, the $MCF_{fx}$ was significantly higher than at 0.15 m, and the $MCF_{fy}$ values at 0.60 m and 0.75 m were significantly higher than those at 0.15 m and 0.30 m.

Finally, muscle force was also significantly affected by landing height. At a height of 0.75 m, the force of the VI was significantly greater than at 0.15 m (p = 0.03), and the forces of the VL at heights of 0.45 m, 0.60 m, and 0.75 m were significantly higher than that at 0.15 m (p = 0.04), and also significantly higher at 0.60 m and 0.75 m compared with 0.30 m. Additionally, the effect of different landing heights on the muscle force of the VM was significant (p = 0.03), with the force at 0.15 m, 0.30 m, 0.45 m, and 0.75 m being significantly greater than that at 0.60 m. The univariate ANOVA results for the effects of different landing heights on the KRM, BFlh, BFsh, RF, SM, and ST were not significant (p>0.05).

The SPM1d results for the F statistic depicted in Figs 3 and 4 indicated that throughout the entire movement process, different landing heights significantly affected the vGRF. Specifically, significant differences in the vGRF were observed during the time interval percentages of 36% to 39%, 46% exactly, and 49.5% to 55.5% (α = 0.05, F = 6.249, p<0.01). The SPM1d F statistic results for the knee joint KFM, KAM, and KRM showed no significant differences (p>0.05).

Further analysis by SPM1d assessing the F statistic (Fig 5) indicated that the $MCF_{fx}$ and $MCF_{fz}$ were also significantly influenced by the landing height. Significant differences in the $MCF_{fx}$ were found in the time interval percentage of 17% to 22% (α = 0.05, F = 5.474, p<0.01),

**Table 1. Comprehensive statistical analysis of knee joint characteristic indicators during different drop jump landing heights (N = 16).**

| Measure | Drop Jump Landing Height | | | | | F | P | $\eta^2$ |
|---|---|---|---|---|---|---|---|---|
| | 0.15 m | 0.30 m | 0.45 m | 0.60 m | 0.75 m | | | |
| | Mean ± SD | Mean ± SD | Mean ± SD | Mean ± SD | Mean ± SD | | | |
| Knee angle (degrees) | -76.36±6.89 | -75.18±4.27 | -72.17±4.34 | -72.37±3.13 | -74.32±10.79 | 0.696 | 0.599 | 0.065 |
| GRF (N/BW) | 2.62±1.06[bcde] | 2.7±1.15[acde] | 3.68±1.27[ab] | 3.93±1.25[ab] | 5.27±1.35[ab] | 31.32 | 0.007 | 0.76 |
| KFM (N/kg) | -2.42±0.01[cde] | -2.19±0.01[cde] | -3.14±0.01[ab] | -3.82±0.02[ab] | -3.77±0.01[ab] | 16.42 | 0.001 | 0.62 |
| KAM (N/kg) | 0.53±0.18[cde] | 0.55±0.23[cde] | 0.99±0.7[ab] | 1.06±0.70[ab] | 1.3±0.96[ab] | 44.94 | 0.00 | 0.83 |
| KRM (N/kg) | 1.09±0.30 | 1.13±0.32 | 1.2±0.42 | 1.22±0.45 | 1.09±0.53 | 6.429 | 0.15 | 0.40 |
| $MCF_{fx}$ (N/BW) | 0.43±0.17[d] | 0.47±0.14 | 0.57±0.18 | 1.17±0.19[a] | 1.14±0.21 | 4.86 | 0.01 | 0.35 |
| $MCF_{fy}$ (N/BW) | -7.33±2.39[de] | -7.84±2.62[de] | -15.48±3.66 | -18.46±4.13[ab] | -17.34±3.94[ab] | 10.44 | 0.00 | 0.52 |
| $MCF_{fz}$ (N/BW) | -5.27±1.7[de] | -5.11±1.67[de] | -6.05±1.92 | -8.06±1.99[ab] | -6.42±1.73[ab] | 8.40 | 0.00 | 0.46 |
| BFlh (N/BW) | 1.35±0.44 | 1.41±0.46 | 1.42±0.47 | 1.36±0.46 | 1.19±0.46 | 0.93 | 0.46 | 0.09 |
| BFsh (N/BW) | 1.06±0.29 | 1.09±0.26 | 0.7±0.2 | 0.49±0.15 | 0.61±0.15 | 2.03 | 0.11 | 0.19 |
| RF (N/BW) | 0.34±0.12 | 0.35±0.12 | 0.36±0.13 | 0.34±0.13 | 0.36±0.12 | 1.46 | 0.23 | 0.14 |
| SM (N/BW) | 1.41±0.43 | 1.48±0.45 | 1.27±0.41 | 1.14±0.36 | 1.15±0.38 | 2.01 | 0.11 | 0.18 |
| ST (N/BW) | 0.86±0.28 | 0.78±0.26 | 0.73±0.25 | 0.86±0.29 | 0.77±0.26 | 0.81 | 0.53 | 0.08 |
| VI (N/BW) | 0.31±0.06[e] | 0.83±0.10 | 1.33±0.20 | 1.06±0.17 | 2.00±0.29[a] | 2.93 | 0.03 | 0.24 |
| VL (N/BW) | 0.23±0.03[cde] | 0.54±0.15[de] | 0.77±0.26[a] | 1.08±0.37[ab] | 1.22±0.40[ab] | 7.29 | 0.04 | 0.45 |
| VM (N/BW) | 0.81±0.22[d] | 0.81±0.24[d] | 0.92±0.27[d] | 0.64±0.21[abce] | 0.84±0.25[d] | 2.99 | 0.03 | 0.25 |

Notes: (1) Significance comparisons between different groups are denoted by letters a-e: "a" indicates a significant difference compared with the 0.15-m group; "b," compared with the 0.30-m group; "c," compared with the 0.45-m group; "d," compared with the 0.60-m group; and "e," compared with the 0.75-m group. (2) The units for joint moment are Nm/kg, standardized based on the participant's body weight. GRF, MCF, and muscle force are all standardized in N/BW (body weight) units. (3) The values defining the effect size indicator $\eta^2$ are as follows: 0.01 represents a small effect; 0.06, a moderate effect; and 0.14, a large effect.

and significant differences in the $MCF_{fz}$ were observed in the time interval percentage of 34% to 36% ($\alpha$ = 0.05, F = 5.455, p = 0.032). The analysis for the $MCF_{fy}$ showed no significant differences (p>0.05).

Finally, the SPM1d F statistic results indicated that different landing heights had a significant impact on the BFsh muscle during the time intervals of 16% to 21%, 44% to 46%, and 81% to 87.5% ($\alpha$ = 0.05, F = 5.802) (p<0.01, p = 0.026, p<0.01, respectively) (Fig 6). There was a significant difference in the muscle force of the VL during the time interval percentage of

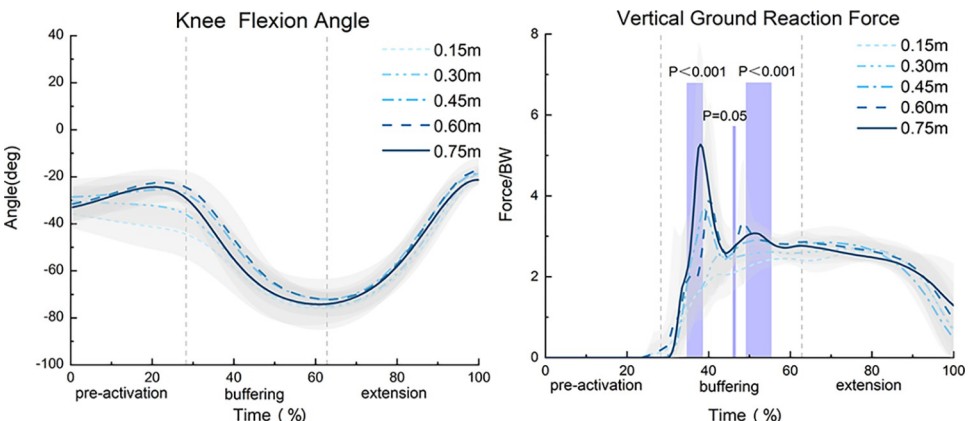

**Fig 3. SPM1d analysis of the knee joint flexion angle and vertical ground reaction force during different drop jump landing heights (N = 16).**

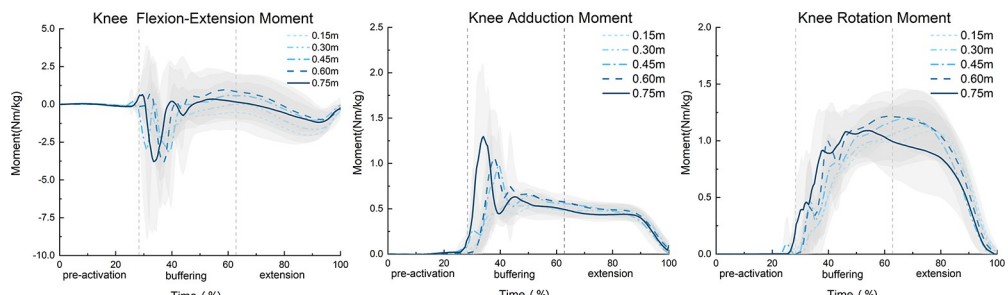

**Fig 4. SPM1d analysis of the knee joint moment throughout different drop jump landing heights (N = 16).**

91% to 92% ($\alpha = 0.05$, F = 5.972, p<0.01). Also, significant changes in the muscle force of the VM were observed during the time interval percentages of 34% to 36% and 41% to 42% ($\alpha = 0.05$, F = 5.784) (p = 0.048, p = 0.045, respectively). The SPM1d F statistic results for the BFlh, RF, SM, and ST showed no significant differences (p>0.05).

## 4 Discussion

The aim of this study was to investigate the tibiofemoral joint MCF and muscular effort during DJs from different landing heights. Specifically, the study aimed to evaluate the impact of different DJ landing heights on the KFM, KAM, KRM, and vGRF, and the characteristics of the surrounding muscular force of the knee.

### 4.1 Impact of different landing heights on joint moment and MCF

Previous studies have indicated that higher drop landing heights significantly increase the peak KFM and KAM during DJs [30]. The study found that the KFM and KAM at DJ landing heights of 0.45 m, 0.60 m, and 0.75 m were significantly higher than those at landing heights of 0.15 m and 0.30 m, supporting our hypotheses and is consistent with previous findings [16, 17]. Fig 4 shows that after landing, the KFM, KAM, and KRM loads increased significantly. Higher drop landing heights convert more potential energy into kinetic energy, increasing the impact force at landing. In single-leg drop Jumps, as the primary site for absorbing and dispersing forces, the knee joint must increase joint moments to cope with this increased impact force. The measured vGRFs confirmed this trend, with a significantly increase observed at higher DJ landing heights, resulting in a greater impact force on the body [31]. Table 1 indicates that for DJs from heights below 0.45 m, although the loads of knee joint moment and vGRF increased with landing height, this increase is minimal, indicating that lower landing

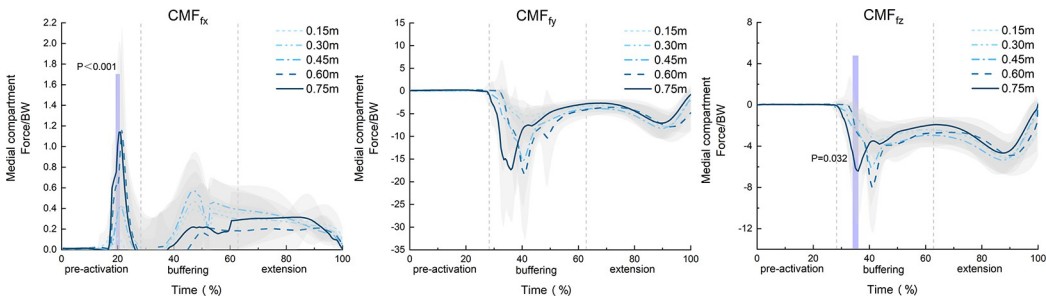

**Fig 5. SPM1d analysis of the contact force in the medial tibiofemoral compartment during different drop jump landing heights (N = 16).**

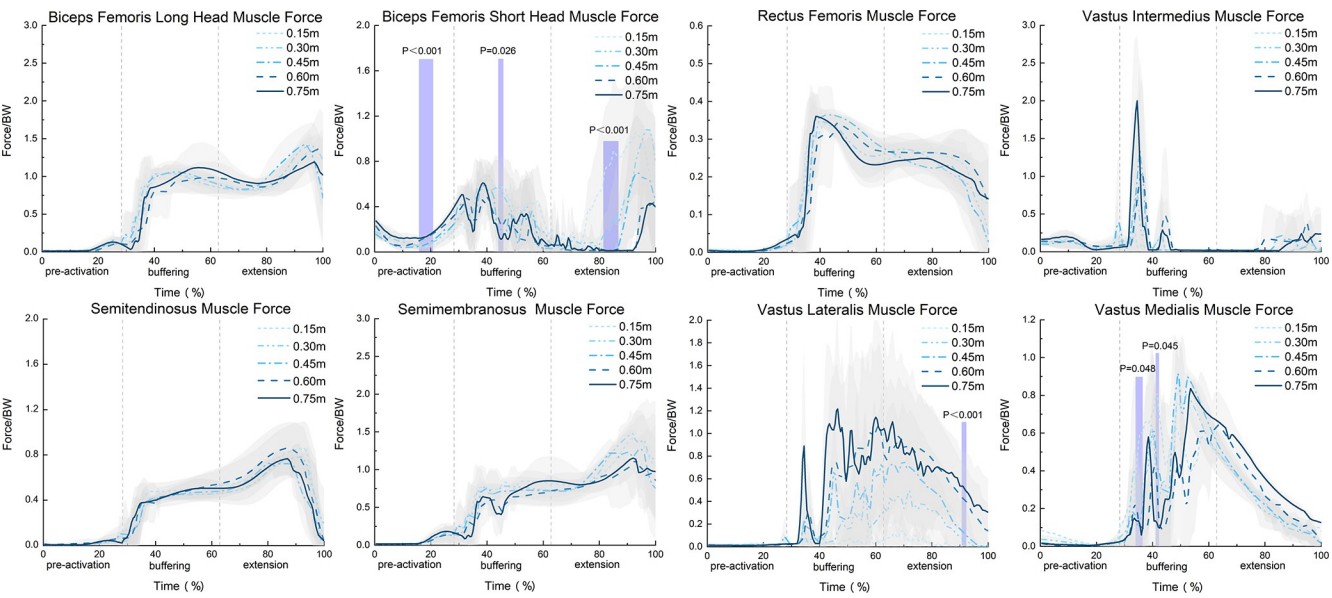

**Fig 6. SPM1d analysis of the peri-knee muscular forces throughout different drop jump landing heights (N = 16).**

height jumps do not significantly increase the load risk on the knee joint [17]. Fig 5 displays the MCF variation curves for five different DJ landing heights, presenting the following three characteristics: (1) At the onset of the landing buffering phase, the medial compartment load surged rapidly, and then gradually with knee flexion. It slightly rose during the push-off phase and dropped again at takeoff. (2) The MCF values increased sequentially in the vertical, internal-external, and front-back directions. (3) Upon landing, the mechanical properties of the knee joint contact forces changed drastically due to the rapid deceleration of the lower limbs, resulting in a sudden increase in GRF and a sharp rise in knee joint load [32]. These characteristics indicated that the medial compartment, especially in the $MCF_{fy}$, experienced a large and rapid load after landing. Related studies on walking, running, and lateral movements have also indicated larger medial compartment contact forces in the $MCF_{fy}$ [18], which aligns with the observations of this study. Our results are also applicable to jumping sports. Hence, the medial compartment of the knee after a DJ will bear a significant load in a short time, especially in the $MCF_{fy}$.

The SPM1d analysis indicated that in the pre-activation phase of the DJ movement, the $MCF_{fx}$ exhibited greater sensitivity to variations to changes in landing height. As the landing height increased, the $MCF_{fx}$ intensified, likely due to the forward shift in the body's center of gravity caused by the higher drop height. Additionally, the phenomenon of muscle pre-activation may also play a role, which is observed in movements such as jumping and walking. Muscle pre-activation refers to the central nervous system activates muscles in advance to prepare for the post-landing movement [33], This mechanism can trigger protective muscle strategies in unforeseen circumstances, thereby reducing the potential for injury. Helm et al.'s research indicates that during the descent phase of the DJ movement, a higher jump height results in increased pre-activation of lower limb muscle, thereby generating stronger muscle force before landing [31]. During the buffering phase of the DJ movement, our study revealed that at a 0.60-m jump height, the $MCF_{fz}$ after landing was significantly higher than that at 0.15 m. Furthermore, at 0.60 m, the peak $MCF_{fz}$ values occurred earlier than at other heights. This could be related to the increased falling speed at higher jump heights, leading to greater impact forces

at landing and possibly causing earlier peak contact forces [34]. Therefore, it is recommended to gradually increase jump heights in high-intensity jump training to reduce the load on the knee joint and prevent potential injuries. Simultaneously, training should strengthen muscle pre-activation and balance coordination abilities to adapt to the higher demands of the DJ movements.

By comparing the changes in the MCF and knee joint moments during DJ movements, we observed that, despite the opposite directions in the change curves for the KFM vs. $MCF_{fx}$, KAM vs. $MCF_{fy}$, and KRM vs. $MCF_{fz}$, the trends were similar throughout the DJ process. This indicates that during complex movements, the knee joint components work in coordination to maintain stability and efficiency [35]. This coordination pattern likely reflects the adaptive movement strategy adopted by the body during DJs, with different types of forces applied in various directions of the knee joint to adapt and control the movement [36]. Therefore, analyzing the MCF offers a new perspective for understanding of knee joint movement mechanisms. Such analyses not only helped to better understand the performance of the knee joint in the present study but also have significant implications for ports training, injury prevention, and rehabilitation.

## 4.2 Impact of different DJ landing heights on the muscular force around the knee joint

Muscular coordination is crucial in determining the load on the tibiofemoral joint [37]. Previous studies indicate that during lateral cutting steps, the quadriceps contribute most significantly to the forces on the medial region [38]. This study explored the characteristics of muscular forces during the DJ movement, founding that the VI, VL, BFlh, and SM, components of the quadriceps, were the main sources of knee joint force. During the buffering phase of DJ landing, rapid quadriceps contraction absorbed the impact force and evenly distributed the load across the knee joint by pulling the tibia upward [39]. The study also found a significant increase in the force of the VI and VL with higher jump heights. For the VL especially, its force increased with height, showing adaptability to higher impacts. Conversely, a significant increase in the force of the VI was mainly observed within the height range of 0.15 m to 0.75 m, suggesting better adaptability to medium heights [40].

During the DJ pre-activation phase, increased landing height significantly boosted the muscular force of the BF short head (BFsh). BFsh stabilizes the knee joint by contracting to mitigate the impact force during landing and to protect tendons from excessive shock. This finding aligns with recent findings by Ghatak et al. on the role of the BF during the gait swing phase [41]. The study found that during the DJ push-off phase, increased landing height significantly reduced the muscular force of the VM, consistent with earlier findings The VM primarily stabilizes the medial side of the knee joint, and its reduced force may lead to decreased medial stability, increased contact forces on the medial joint, and raise the risk of knee injuries, particularly to the medial collateral ligament or knee valgus [42]. Thus, boosting the VM's muscular force is vital to lowering the risk of knee injuries. Training should focus on strengthening the VM during this movement phase to improve knee joint stability and minimize injury risk.

In the DJ push-off phase, higher landing heights significantly increase VL muscular force This increased force generates a higher moment and distribute joint contact forces more evenly, alleviating excessive pressure in certain areas [43].

The study found that at a landing height of 0.75 m, the knee joint load was lower than at 0.60 m, despite higher muscular force at 0.75 m. Previous research indicates that higher DJ jump heights may stimulate the neuromuscular system, prompting muscles to adapt to reduce

joint loads. Faced with higher jumps, athletes may pre-activate their muscles to manage greater impact forces. This pre-activation aids in controlling knee joint motion and stability, thus reducing local loads [44]. Furthermore, as the DJ height increases, athlete s ' motor system may adopt a more protective landing strategy due to the perceived higher risk. Increased height results in mechanical changes, including longer air time, which allows for better body position adjustments and force distribution [45].

### 4.3 Limiting factors and future research

This study examined the impact of different DJ landing heights on the knee joint mechanical. However, it has several limitations. Firstly, none of the participants were female athletes. Given that multiple studies have shown differences in the lower limb biomechanics between females and males in DJs, future research should include female participants to explore gender impact [46]. Secondly, this study did not assess the coordination between the lower limb joints. This coordination is crucial for completing the movement and reducing the lower limb injury risk. Future research should include assessments of the coordination and the impact of muscle activation on it. Thirdly, Verniba et al. found that increased height significantly enhances the total contact force in the vertical direction of the knee joint during the push-off phase of DJs [17]. Therefore, future research should explore the load transfer methods between the knee joint compartments and methods to measure muscular force contributions to contact forces in the medial and lateral compartments [45].

## 5 Conclusion

In this study, we found that increased landing heights significantly augmented medial compartment forces (MCF) and influenced knee joint moments (KFM, KAM, and KRM) in patterns similar to the changes in MCF components (MCFfx, MCFfy, and MCFfz) throughout the entire DJ process. This suggests that the knee joint segments effectively coordinate to maintain stability and movement efficiency during complex actions like DJs, representing an adaptive movement strategy. Furthermore, the impact of different landing heights on major muscle groups around the knee, such as the quadriceps, was evident. Higher jump heights notably increased the strength of the VI and VL, while the strength of the VM decreased. These findings highlight the need for a gradual increase in jump height combined with reinforced muscle pre-activation and balance coordination training to mitigate the risk of knee injuries. This study contributes to the existing body of knowledge by emphasizing the importance of controlled jump training to enhance knee joint stability and prevent injuries, which has significant implications for athletic training and rehabilitation programs.

## Supporting information

**S1 Dataset. All data includes raw data for all figures related to this article.** It is available at https://doi.org/10.6084/m9.figshare.25953808.v3.
(XLSX)

## Acknowledgments

The authors thank Jianxin Du and Shougui Luo for assistance in data collection and for comments on early versions of this paper.

## Author Contributions

**Conceptualization:** Gengchao Bi, Lijun Hua, Jiajie Sun, Qiang Xu.

**Data curation:** Gengchao Bi, Qiang Xu, Guanbo Li.

**Formal analysis:** Gengchao Bi, Qiang Xu.

**Funding acquisition:** Lijun Hua.

**Investigation:** Gengchao Bi, Qiang Xu, Guanbo Li.

**Methodology:** Gengchao Bi, Qiang Xu, Guanbo Li.

**Project administration:** Lijun Hua.

**Software:** Gengchao Bi, Qiang Xu, Guanbo Li.

**Supervision:** Gengchao Bi, Lijun Hua.

**Validation:** Gengchao Bi.

**Visualization:** Gengchao Bi, Jiajie Sun.

**Writing – original draft:** Gengchao Bi, Lijun Hua.

**Writing – review & editing:** Gengchao Bi, Lijun Hua.

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
