## [Decision Letter · Decision Letter 0]

20 May 2024

PONE-D-24-13422Impact of different landing heights on the contact force in the medial tibiofemoral compartment and the surrounding muscle force characteristics in drop jumpsPLOS ONE

Dear Dr. Hua,

Thank you for submitting your manuscript to PLOS ONE. After careful consideration, we feel that it has merit but does not fully meet PLOS ONE’s publication criteria as it currently stands. Therefore, we invite you to submit a revised version of the manuscript that addresses the points raised during the review process.

We look forward to receiving your revised manuscript.

Kind regards,

Mehrnaz Kajbafvala, Ph.D

Academic Editor

PLOS ONE

2. We note that Figure 2 in your submission contain copyrighted images. All PLOS content is published under the Creative Commons Attribution License (CC BY 4.0), which means that the manuscript, images, and Supporting Information files will be freely available online, and any third party is permitted to access, download, copy, distribute, and use these materials in any way, even commercially, with proper attribution. For more information, see our copyright guidelines: http://journals.plos.org/plosone/s/licenses-and-copyright.

1. You may seek permission from the original copyright holder of Figure 2 to publish the content specifically under the CC BY 4.0 license.

Reviewers' comments:

Reviewer's Responses to Questions

**Comments to the Author**

1. Is the manuscript technically sound, and do the data support the conclusions?

Reviewer #1: Yes

Reviewer #2: Yes

2. Has the statistical analysis been performed appropriately and rigorously? 

Reviewer #1: I Don't Know

Reviewer #2: Yes

3. Have the authors made all data underlying the findings in their manuscript fully available?

Reviewer #1: Yes

Reviewer #2: Yes

4. Is the manuscript presented in an intelligible fashion and written in standard English?

Reviewer #1: Yes

Reviewer #2: Yes

5. Review Comments to the Author

Reviewer #1: This article investigates Impact of different landing heights on the contact force in the medial tibiofemoral compartment and the surrounding muscle force characteristics in drop jumps.

The statistical analysis has been performed appropriately and the manuscript written in standard English.

The comments below may help you rewrite the article.

Abstract:

- The conclusion is not clearly stated. In my opinion, it is better to state the conclusion of the study instead of practical suggestions for the study.

Main text

Introduction:

The problem statement should be revised. I did not understand what is the main difference between your study and previous studies.

- Why did you insert 16 participants? How was the sample size calculated?

Subject Selection:

"We recruited 16 male participants with 3-5 years of sports training experience……." What types of athletes did you include? Does the difference in the type of exercise interfere with the results of the study?line 123: insert refence of by Delp et al.

Results:

- lines 245-248: Specifically, the vGRF at a height of 0.15 m was significantly lower than that at other heights……… Additionally, the vGR increased significantly with landing height.

These two sentences have a repeated meaning. I think you can delete one.

Discussion:

The discussion section is too long. I think it is better to rewrite briefly and comprehensively.

Reviewer #2: Your manuscript is very well written and i hope you good luck in future research, in the method part i made some recommendations for you and i think it would be better to separate athletes based on their specific goals and fields

Please clarify in conclusion part a comparison between legs about force transmission

With regards

6. PLOS authors have the option to publish the peer review history of their article (what does this mean?). If published, this will include your full peer review and any attached files.

Reviewer #1: No

Reviewer #2: **Yes: **Pezhman Masoudi

---

## [Author Response · Author response to Decision Letter 0]

5 Jun 2024

Response to the comments of Reviewers 

Response to the comments of Reviewer 1

Dear Reviewer,

We deeply appreciate your meticulous review of our paper. Following your suggestions, we have revised the relevant sections and made certain changes to the manuscript. These alterations do not affect the core content or overall structure of the paper. We are profoundly grateful for your professionalism and dedication, and hope that the revised paper gains your approval. All changes are detailed in the following "Response" section and the revised content is highlighted in yellow in the original manuscript for easy reference. We hope our revised manuscript will be acceptable to you.

We look forward to your valuable feedback and suggestions on our revised manuscript. We sincerely appreciate the time and effort invested by the reviewer in evaluating our paper and look forward to receiving your additional feedback or suggestions.

Sincerely, 

Lijun Hua

In the subsequent section, a thorough response is offered in relation to the comments you have provided.

Comments:

(1)Abstract 

The conclusion is not clearly stated. In my opinion, it is better to state the conclusion of the study instead of practical suggestions for the study”

Response:

Thank you for providing valuable feedback. We have incorporated the revisions into the conclusion of the "Abstract" section of the manuscript. The specific modifications are as follows: 

1) Stating research findings: During drop jumps, both the knee moment and the MCF exhibited coordinated changes. 

2) Removal of practical suggestions: We have deleted the provided practical operational suggestions and focused on the research conclusions themselves. 

3) Highlighting the significance of the study: Emphasis has been placed on the impact of different drop jump heights on the muscles around the knee joint, as well as the specific muscular responses and their importance for injury prevention.

The revised conclusion in the abstract is as follows: 

"Thus, both the knee moment and the MCF exhibited similar coordinated changes during drop jumps, indicating that these may be adaptive movement strategy. The impact of varying drop jump heights on muscle groups around the knee joint varied suggest that different heights induce specific muscular responses and improve muscle coordination to prevent knee joint injuries."

#Please note that the specific revisions mentioned can be found in the "Manuscript" file, lines 28-32. 

#Alternatively, you can refer to the "Revised Manuscript with Track Changes" file , where the indicated lines 28-33 are highlighted with a yellow background by the author.

Comments

(2):Main text Introduction

The problem statement should be revised. I did not understand what is the main difference between your study and previous studies.

Response:

We appreciate your valuable feedback and in-depth insights. Your suggestion to highlight the differences between our research and previous studies is highly constructive. We agree to make the revisions and have incorporated them into the Introduction section.

Specific modifications are as follows:

1) Emphasizing the differences between our research and previous studies: It is explicitly pointed out that there is a scarcity of research on the tibiofemoral joint's medial compartment force (MCF) during jump landing movements. Previous studies have relied predominantly on the knee adduction moment (KAM).

The specific revisions are as follows:

"Despite its importance, research specifically focusing on the tibiofemoral joint's medial compartment force (MCF) during DJ movements is scarce. Previous studies have often relied on the KAM as a proxy, which does not fully capture the load distribution characteristics of the medial compartment."

#Please refer to the specific modifications in the "Manuscript" file, lines 64-68. #Alternatively, you can find the exact changes in the "Revised Manuscript with Track Changes" file, in the "No Markup" mode, where the highlighted sections can be found in lines 67-70 with a yellow background.

2) Strengthening the research objectives: This study aims to provide a more precise understanding of knee joint loading by analyzing the impact of different landing heights on MCF, KAM, and muscle characteristics. The goal is to offer guidance for athletes in selecting appropriate jump landing heights.

The specific revisions are as follows:

"By analyzing the impact of different landing heights on these characteristics ,this study aims to provide a more precise understanding of knee joint loading and the role of muscle forces during DJs. This will offer valuable insights for athletes in selecting suitable DJ landing heights to minimize knee joint load, enhancing current knowledge on DJ training and its implications for knee joint health. "

#Please refer to the specific modifications in the "Manuscript" file, lines 93-97. #Alternatively, you can refer to the "Revised Manuscript with Track Changes" file , where the indicated lines 100-106 are highlighted with a yellow background by the author.

Comments: 

(3)Subject Selection: Why did you insert 16 participants? How was the sample size calculated?

Response:

Thank you for providing thoughtful and constructive feedback. Regarding the calculation of sample size, we used GPower to estimate the sample size and consulted relevant literature (as listed below). The number of participants ranged from approximately 10 to 16. However, we did not provide detailed explanations in the manuscript. We appreciate your attentive reminder and suggestion. We have decided to accept the revision and will add the method for calculating the sample size in the article.

The specific revisions are as follows:

"A prediction for the sample size was carried out using GPower 3.1.9.7, with a significance level (α) of 0.05 and a statistical power of 80%. The analysis determined that a minimum sample size required was 13 individuals. "

#Please refer to the specific modifications in the "Manuscript" file, lines 106-108. #Alternatively, you can refer to the "Revised Manuscript with Track Changes" file , where the indicated lines 117-119 are highlighted with a yellow background by the author.

Below are the references:

Peng, H. T. (2011). Changes in biomechanical properties during drop jumps of incremental height. The Journal of Strength & Conditioning Research, 25(9), 2510-2518.(This article also selected 16 young, healthy college students.)

Sinclair J, Hobbs SJ, Selfe J. The Influence of Minimalist Footwear on Knee and Ankle Load during Depth Jumping. Research in Sports Medicine. 2015;23(3):289-301. doi: 10.1080/15438627.2015.1040917.(This article selected 10 male participants.)

Tsatalas, T., Karampina, E., Mina, M. A., Patikas, D. A., Laschou, V. C., Pappas, A., ... & Giakas, G. (2021). Altered drop jump landing biomechanics following eccentric exercise-induced muscle damage. Sports, 9(2), 24.(This article selected 15 regional level athletes.)

Comments:

(4)"We recruited 16 male participants with 3-5 years of sports training experience……." What types of athletes did you include? Does the difference in the type of exercise interfere with the results of the study?

Response:

We apologize for overlooking the participants' types of sports and whether it influenced the research results. We appreciate the reviewer's suggestion and have made revisions accordingly. Our participants primarily come from sports such as basketball, volleyball, and soccer, which involve frequent jumping and landing movements similar to those in our study. This ensures the representativeness and consistency of the research findings. Furthermore, we have added relevant content to the manuscript.

The specific revisions are as follows:

"The participants predominantly engaged in sports such as basketball, volleyball, and soccer, which necessitate frequent jumping and landing movements analogous to the drop jumps executed in this study. " 

#Please refer to the specific modifications in the "Manuscript" file, lines 109-111. #Alternatively, you can refer to the "Revised Manuscript with Track Changes" file , where the indicated lines 120-122 are highlighted with a yellow background by the author.

Comments:

(5)line 123: insert refence of by Delp et al.

Response:

Thank you for your careful review. We apologize for the oversight in missing references in the article. We sincerely apologize for the mistake. We have taken your suggestion into account and have added the reference by Delp et al.

The specific reference is as follows:

Delp SL, Anderson FC, Arnold AS, Loan P, Habib A, John CT, et al. OpenSim: open-source software to create and analyze dynamic simulations of movement. IEEE Trans Biomed Eng. 2007;54(11):1940-50. Epub 2007/11/21. doi: 10.1109/TBME.2007.901024. PubMed PMID: 18018689.

#Please refer to the specific modification at Line 130 in the "Manuscript" file. #Alternatively, you can find the exact modification location in the "Revised Manuscript with Track Changes" file at line 142. The author has highlighted it with a yellow background for better visibility.

Comments:

(6)Results:

 lines 245-248: Specifically, the vGRF at a height of 0.15 m was significantly lower than that at other heights. Additionally, the vGRF increased significantly with landing height.These two sentences have a repeated meaning. I think you can delete one.

Response:

Thank you for your careful review. After consideration, we have decided to remove the duplicate sentence in order to avoid repetition. Thank you again for your diligent and thorough review of the manuscript.

#For the specific modification location, please refer to Line 248 in the "Manuscript" file, where it has already been deleted.

#Alternatively, you can refer to the "Revised Manuscript with Track Changes" file , where the indicated lines 263-264 .

Comments:

(7)Discussion:

The discussion section is too long. I think it is better to rewrite briefly and comprehensively.

Response:

Thank you for your valuable suggestions. We have realized that the discussion section is somewhat bulky, so based on your feedback, we have made modifications and deletions to some sentences in order to refine the discussion while maintaining the overall structure. We will further streamline the discussion to ensure that all content is closely tied to the main theme and is concise. Once again, we appreciate your patient review and valuable suggestions.

Regarding your feedback, let me briefly explain the modifications made to the "Discussion" section in the manuscript.

Firstly, we have eliminated parts of the discussion section that contain semantic repetition. This mainly includes: 

Reduction of the simple understanding and comparison of the data in Table 1 in the manuscript. - Partial removal of excessive descriptions of the body's center of gravity in the text. 

Deletion of repetitive conclusions and viewpoints in the manuscript. 

Reduction of redundant statements about the limitations of the study in the manuscript. As well as some other deletions in specific parts.

#Please refer to the "Revised Manuscript with Track Changes" file for details of the deleted content in the discussion section.

Secondly, in the discussion section, each sentence has been refined and rewritten to make it more concise and fluent. For specific modification content and location, #Please refer to the "Revised Manuscript with Track Changes" file for details of the discussion section . The author has highlighted the modifications with a yellow background.

Response to the comments of Reviewer 2

Dear Reviewer,

We would like to extend our sincere gratitude for your thorough evaluation of our manuscript. In response to your insightful recommendations, we have carefully revised the pertinent sections while ensuring that the core content and overall structure of the paper remain unchanged. We greatly value your expertise and commitment, and we are hopeful that the revised paper meets with your approval. Detailed information regarding all modifications is provided in the "Response" section, and the revised content is highlighted in yellow within the original manuscript for your convenience. We are optimistic that our revised manuscript aligns with your expectations.

Your expertise and dedication to the review process are deeply appreciated, and we sincerely hope that the revised manuscript is in line with your expectations.

Sincerely, 

Lijun Hua

In the subsequent section, a thorough response is offered in relation to the comments you have provided.

Comments:

(1)Abstract 

Please replace the word strengthen with improve ‘strengthen muscle coordination’ in the last line.

Response:

 Thank you for the valuable suggestion. We accept this revision and will replace "strengthen" with "improve" .

"The impact of varying drop jump heights on muscle groups around the knee joint varied suggests that different heights induce specific muscular responses and improve muscle coordination to prevent knee joint injuries."

#The specific modifications are located in "Manuscript" file at Line30-32.

#Alternatively, you can refer to the "Revised Manuscript with Track Changes" file , where the indicated lines 30-33 are highlighted with a yellow background by the author.

Comments:

(2)Introduction

Line 85 please correct "leading a decrease in knee stiffness’

At higher drop jump heights, the increased impact forces can lead to changes in muscle activation patterns, resulting in a decrease in knee stiffness.

Response:

Thank you for the valuable comments from the reviewer. Based on your suggestions, we have made revisions. 

The specific changes are as follows:

"At higher drop jump heights, the increased impact forces can lead to changes in muscle activation patterns, resulting in a decrease in knee stiffness."

#For the specific changes, please refer to Lines 84-85 in the "Manuscript" file. #Alternatively, you can refer to the "Revised Manuscript with Track Changes" file , where the indicated lines 91-93 are highlighted with a yellow background by the author.

Comments:

(3)Methods

Why did you recruit only male subjects, as many knee disorders such as PFA or PFPS is more common in female athletes?

Response: 

Dear reviewer, thank you for your valuable suggestion. Regarding your concerns, we completely understand the rationale behind your question. The reason for recruiting only male subjects was to control for the variability in biomechanical responses related to gender, ensuring the accuracy of our research findings. We are aware that many knee disorders, such as patellofemoral pain syndrome (PFPS) and patellofemoral arthritis (PFA), are more common in female athletes. In our future studies, we will definitely include female subjects to addressing this limitation and gaining a comprehensive understanding of the biomechanics of the knee joint across different genders. This aspect has also been addressed in line 428 of the manuscript. Once again, thank you for your guidance and attention. We will strive to further improve our research.

Comments:

(4)Were all the athletes engaging in one specific sport stream or different sports, please clarify, as different sport athletes will show different biomechanics of knee joint.

Response:

Dear reviewer, thank you very much for your query and attention. Regarding your question, we are pleased to clarify that the athletes recruited for this study were from various sports including soccer, basketball, and track and field, among others. This diversity in selection aimed to capture a range of biomechanical responses associated with different sports. In response to your suggestion, we have included the statement "The participants predominantly engaged in sports such as basketball, volleyball, and soccer, which necessitate frequent jumping and landing movements analogous to the drop jumps executed in this study" in the manuscript to adequately address your query. Once again, thank you for your guidance, and we will strive to further enhance the quality of our research.

#The specific modification has been made in lines 109 to 111 of the manuscript. 

#Alternatively, you can refer to the "Revised Manuscript with Track Changes" file , where the indicated lines 120-122 are highlighted wi

---

## [Decision Letter · Decision Letter 1]

8 Jul 2024

Impact of different landing heights on the contact force in the medial tibiofemoral compartment and the surrounding muscle force characteristics in drop jumps

PONE-D-24-13422R1

Dear Dr. Lijun Hua,

We’re pleased to inform you that your manuscript has been judged scientifically suitable for publication and will be formally accepted for publication once it meets all outstanding technical requirements.

Kind regards,

Mehrnaz Kajbafvala, Ph.D

Academic Editor

PLOS ONE

Additional Editor Comments (optional):

Reviewers' comments:

Reviewer's Responses to Questions

**Comments to the Author**

1. If the authors have adequately addressed your comments raised in a previous round of review and you feel that this manuscript is now acceptable for publication, you may indicate that here to bypass the “Comments to the Author” section, enter your conflict of interest statement in the “Confidential to Editor” section, and submit your "Accept" recommendation.

Reviewer #2: All comments have been addressed

Reviewer #3: All comments have been addressed

2. Is the manuscript technically sound, and do the data support the conclusions?

Reviewer #2: Yes

Reviewer #3: Yes

3. Has the statistical analysis been performed appropriately and rigorously? 

Reviewer #2: Yes

Reviewer #3: Yes

4. Have the authors made all data underlying the findings in their manuscript fully available?

Reviewer #2: No

Reviewer #3: Yes

5. Is the manuscript presented in an intelligible fashion and written in standard English?

Reviewer #2: Yes

Reviewer #3: Yes

6. Review Comments to the Author

Reviewer #2: The article is very well written and revised in almost all comments, the revised version is more structured and straightforward, wish you good luck in future researches.

Reviewer #3: Congratulation

Your answers are acceptable......................................................................................................................................

..............................

7. PLOS authors have the option to publish the peer review history of their article (what does this mean?). If published, this will include your full peer review and any attached files.

Reviewer #2: **Yes: **Pezhman masoudi PT PhD

Reviewer #3: **Yes: **Soheil Mansour Sohani

---

## [Editor Report · Acceptance letter]

11 Jul 2024

PONE-D-24-13422R1 

PLOS ONE

Dear Dr. Hua, 

I'm pleased to inform you that your manuscript has been deemed suitable for publication in PLOS ONE. Congratulations! Your manuscript is now being handed over to our production team.

Kind regards, 

on behalf of

Dr. Mehrnaz Kajbafvala 

Academic Editor

PLOS ONE